# Metformin as a Potential Treatment Option for Endometriosis

**DOI:** 10.3390/cancers14030577

**Published:** 2022-01-24

**Authors:** Żaneta Kimber-Trojnar, Dominik Franciszek Dłuski, Magdalena Wierzchowska-Opoka, Monika Ruszała, Bożena Leszczyńska-Gorzelak

**Affiliations:** Chair and Department of Obstetrics and Perinatology, Medical University of Lublin, 20-090 Lublin, Poland; zkimber@poczta.onet.pl (Ż.K.-T.); magdaopoka11@gmail.com (M.W.-O.); monika.ruszala@wp.pl (M.R.); b.leszczynska@umlub.pl (B.L.-G.)

**Keywords:** endometriosis, medical therapy, metformin, inflammation

## Abstract

**Simple Summary:**

The aim of this article is to present current knowledge regarding the possibilities of using metformin in the pharmacological treatment of endometriosis. Metformin is an insulin sensitizer widely used for the treatment of type 2 diabetes mellitus. The pleiotropic effects of metformin are mainly exerted through the activation of AMP-activated protein kinase, which is the key cellular energy homeostasis regulator that inhibits mTOR, a major autophagy suppressor. Metformin regresses endometriotic implants by increasing the activity of superoxide dismutase. It is also an inhibitor of metalloproteinase-2, decreasing the levels of the vascular endothelial growth factor and matrix metalloproteinase-9 in animal studies. With its unique therapeutic mechanisms and no serious side effects, metformin seems to be a helpful anti-inflammatory and anti-proliferative agent in the treatment of endometriosis. It could be a missing link for the successful treatment of this chronic disease.

**Abstract:**

Endometriosis is a common disease in women of reproductive age, and its pathogenesis seems to be largely affected by hormone imbalance, inflammation, oxidative stress, and autophagy dysregulation. These pathophysiological disturbances interact with one another through mechanisms that are still awaiting elucidation. The aim of this article is to present current knowledge regarding the possibilities of using metformin in the pharmacological treatment of endometriosis. Metformin is an insulin sensitizer widely used for the treatment of type 2 diabetes mellitus. The pleiotropic effects of metformin are mainly exerted through the activation of AMP-activated protein kinase, which is the key cellular energy homeostasis regulator that inhibits mTOR, a major autophagy suppressor. Metformin regresses endometriotic implants by increasing the activity of superoxide dismutase. It is also an inhibitor of metalloproteinase-2, decreasing the levels of the vascular endothelial growth factor and matrix metalloproteinase-9 in animal studies. In endometriosis, metformin might modify the stroma–epithelium communication via Wnt2/β-catenin. With its unique therapeutic mechanisms and no serious side effects, metformin seems to be a helpful anti-inflammatory and anti-proliferative agent in the treatment of endometriosis. It could be a missing link for the successful treatment of this chronic disease.

## 1. Introduction

### 1.1. Pathophysiology of Endometriosis

Endometriosis is a multidimensional, chronic and incurable disease of unclear aetiology [1]. Although many mechanisms leading to the development of this condition are still awaiting elucidation, several hypotheses on the pathogenesis of endometriosis, both local and systemic, have been emphasized. According to [2,3], numerous interactions at the hormonal, genetic, immunological, and environmental levels are responsible for the development of this type of pathology.

One of the latest widely presented theories postulates that endometriosis arises from an abnormal response to inflammatory processes [4]. Reaching the pelvis via transtubal retrograde flow, the overreactive uterine endometrium may cause adhesion, implantation, and proliferation of the ectopic tissue into the peritoneal cells [5,6,7]. Some studies demonstrate fundamental changes in the concentration of interleukins (ILs; i.e., IL-10, IL-6, IL-8), growth factors, or adhesion molecules in the peritoneal fluid and peripheral blood obtained from patients suffering from endometriosis [8,9]. Metalloproteinases (MMPs), prostaglandins (PGs), and chemokines may also be released [10,11,12].

The adhesion of endometrial fragments to the peritoneum is enhanced by upregulated biomolecules such as selectins, integrins, and cadherins, whereas the process of the further implantation and remodelling of the cellular matrix is controlled by MMP-1, 2, 9, and 11 [13,14]. A defective immune response has a triggering effect on both cellular and humoral responses. Progesterone, which prevents stimulation of MMP-3, maintains tissue stability during the invasive events of implantation [15,16]. MMP-2 and MMP-9 control cyclooxygenase-2 (COX-2) sensitivity in the endometrial cells [17]. The production of COX-2 depends on hormonal status during the menstrual cycle and increases in the PGE2 level. According to some studies, COX-2 expression is higher in females suffering from endometriosis in comparison to healthy controls [18]. This could explain the survival, migration, and invasion of endometrial cells and the stimulation of PGE2 production [19]. The COX-2/PGE2 pathway increases the production of local oestrogen in the endometrium by stimulating the synthesis of aromatase. The PGE2, PGE3, and PGE4 receptors activate many mechanisms, including the mitogen-activated protein kinase (MAPK), phosphatidylinositol-3 kinase/protein kinase B (PI3K/Akt), and Wingless-Type MMTV Integration Site Family (Wnt) pathways, thereby altering, among others, cell apoptosis, proliferation, invasion, and migration [20,21]. The microenvironment is populated by diverse cell types, including fibroblasts, adipocytes, and a vast range of inflammatory cells. Macrophages are highly activated within the peritoneum of women with endometriotic changes [22,23,24]. They contribute to the destruction, repair, and regeneration of endometrial cells; however, macrophages with a dysfunctional phenotype inhibit phagocytosis [25]. A lack of immune response mechanisms may lead to greater ectopic survival, thereby promoting the growth of endometriosis.

Survival of endometrial fragments outside the uterine cavity and their increased proliferation is possible due to impaired apoptosis [26]. Inhibition of cell death is activated by the PI3K/Akt pathway, which controls Bcl2 expression [27]. Stimulation of B lymphocytes induced by endometriosis foci additionally causes the production of autoantibodies, which has an enhancing effect on the inflammatory process [28]. Moreover, the Akt pathway induces phosphorylation of Bad protein and sequestration of Bad and Bax proteins and, thus, promotes the survival of endometrial cells [29]. The activation of nuclear factor kappa-light-chain-enhancer of activated B cells (*NF*-κB) confers resistance to apoptosis phenotype in endometrial cells. It activates macrophages to release proinflammatory cytokines [30]. Transforming growth factor (TGF) and platelet-derived growth factor (PDGF), both macrophage-derived growth factors, might activate Fas-mediated apoptosis of immune cells, protect the endometrial cells from an attack by T-lymphocytes, and escape from immune surveillance in the peritoneal cavity [31].

PDGF, released by peritoneal macrophages, and vascular endothelial growth factor (VEGF), considered the most crucial angiogenic agent, may launch neovascularization [32,33]. The formation of new capillaries is also reinforced by the induction of the PI3K/Akt pathway without hypoxia [34]. The MMP enzymes, especially MMP-2 and MMP-9, are involved in the blood vessel breakdown and basement membrane degradation. Other factors include IL-8 and VEGFA/C, as well as Hypoxia-inducible factor 1 (HIF-1) [35,36]. Angiopoietin (ANG) 1 and 2 are molecules controlled by the tyrosine kinase with immunoglobulin and EGF homology domains (TIE-2) receptor tyrosine kinase (RTK) [37]. The main role of ANG-2 is the structural destabilization of the vessels. The action of both ANG-2 and VEGF results in the migration and proliferation of endothelial cells [38].

Many scientific studies have emphasized a documented link between oestrogens and endometriosis. Aromatase is an essential enzyme for oestrogen biosynthesis, and its role is to convert C19 steroids to oestrogens [39,40]. Elevated oestrogen levels stimulate epithelial proliferation throughout the female reproductive tract [41]. The proliferation of the endometriotic cells may cause infertility and pain, two of the most characteristic pathogenetic features [42]. Oestrogen itself stimulates COX-2, a mediator of inflammatory response. An overproduction of oestradiol and overexpression of oestrogen receptor β (ERβ) in the endometrial stromal cells inhibit the tumour necrosis factor (TNF)-α-mediated apoptosis. As a result of this process, the endometrial tissue in ectopic locations may persist for a long time. Moreover, oestrogens may induce the secretion of IL-1 and PGE2, which generate and maintain local inflammation [43,44]. Another feature of PGE is increased aromatase activity, normally expressed in the human ovary, skin, adipose tissue, and brain but not observed in the endometrial cells of healthy women. The action of oestradiol depends on the cell types involved during different phases of the disease, the amount of hormones, the immune stimulus, and the microenvironment [45].

The local microbiome and estrobolome, a collection of genes encoding oestrogen-metabolizing enzymes in the gut microbiome, seem to be interesting factors capable of modulating the course of endometriosis [46]. Their potential influence on the gut–brain axis makes endometriosis more complex to diagnose and treat [47].

Despite a vast amount of knowledge about inflammatory response, hormonal, genetic, and environmental factors, their involvement in the pathogenesis of endometriosis remains contradictory and is still debated. None of the presented theories provides a clear-cut explanation of the phenomena engaged in this disease. The aim of this manuscript is to present a medical treatment option for endometriosis based on the use of metformin. A better understanding of the pathways in the pathogenesis of endometriosis could facilitate early and more effective treatment of patients suffering from this disease and improve their quality of life.

### 1.2. Current Treatment Options for Endometriosis

Currently, conventional endometriosis treatments consist of invasive surgical procedures, medications, or both. Recommended pharmacological therapies include pain-relieving, nonsteroidal anti-inflammatory drugs, aromatase inhibitors, progestins, combined hormonal contraceptives, and selective progesterone receptor modulators [1,48,49]. The actual first-line therapy includes oestrogen–progestin contraceptives, subsequently agonists and antagonists of gonadotropin-releasing hormone (GnRH).

The use of progesterone promotes a reduction in inflammatory response. Unfortunately, constant inflammation might cause resistance to progesterone, which can result in an insufficient effect of hormone therapy for a subset of endometriotic patients [50,51]. To avoid this phenomenon, synthetic progestins (dienogest) or a high dose of progestins should be applied. Such a therapy modifies the expression of progesterone receptors and decreases proinflammatory cytokines [52].

A multidisciplinary surgery is suggested in instances of failure of pharmacological therapies, recurrence, or multiorgan localization [53,54].

The current treatment aims to eradicate the symptoms in women with endometriosis. Conventional drugs slow down oestrogen production and merely relieve the symptoms rather than cure the illness [55,56]. Furthermore, their unfavourable side effects, including a birth control action for women seeking pregnancy, price, and progesterone-resistant state, restrict their long-term use [50,57]. Additionally, numerous patients experience disease recurrence following the surgery [55].

It should also be emphasized that the mechanisms of endometriosis are thus far unexplained. The undesirable adverse effects of pharmacotherapy and the recurrence after surgical procedures have caused much scientific attention has been paid to alternative and complementary medicines in the treatment of this disease [58]. The priority should be to find alternative approaches associated with hormone therapy or either used as a monotherapy in patients for whom current treatment is contraindicated, intolerant or inappropriate. Consequently, new medical therapies with limited side effects are necessary.

Therefore, there is an urgent need to explore safe and cheap medicaments, which have been used for many years in other diseases, in treating endometriosis. The purpose of this article is to introduce the current knowledge on the possibilities of using metformin in the pharmacological treatment of endometriosis. Additionally, this review aims to recapitulate the potential actions of metformin in endometriosis by acting on angiogenesis, apoptosis, cellular adhesion, and inflammation (Figure 1).

## 2. Metformin—State of the Art

Metformin is an antidiabetic medicament currently used as the first-choice treatment for type 2 diabetes mellitus (T2DM), which does not cause excessive hypoglycaemia [59]. Metformin leads to a significant reduction in plasma fasting insulin levels and a reduction in insulin resistance. Therefore, it is considered an insulin sensitizer, which seems to be associated with its beneficial effects on tyrosine kinase activity and insulin receptor expression [60]. Metformin, by regulation of multiple components of the incretin axis, may have a positive effect on metabolism. Maida et al. [61] presented that this agent stimulates gene expression of islet incretin receptor by a mechanism that is dependent on peroxisome proliferator-activated receptor (PPAR)-α and intensely raises levels of glucagon-like peptide 1 (GLP-1) in plasma.

Nevertheless, a great deal of argumentation from clinical researchers confirms that the fundamental action of metformin is to reduce intrahepatic glucose production, mostly by inhibiting gluconeogenesis through a mild and temporary inhibition of the mitochondrial respiratory chain complex 1 [62,63,64]. Furthermore, the decline in hepatic energy status initiates the AMP-activated protein kinase (AMPK), which provides a mechanism for metformin function on the hepatic gluconeogenic program.

Metformin also decreases obesity-associated inflammation and other inflammatory reactions and affects the steroidogenesis in ovarian granulosa and thecal cells [65,66,67,68,69,70,71]. Metformin inhibits plasminogen activator inhibitor-1 (PAI-1) levels, endometrial androgens receptor expression, and plasmatic endothelin-I (ET-1), which are factors that increase the risk of miscarriage.

The therapeutic potential of this medicament, which is most likely induced by the improvement in insulin sensitivity, is employed in different conditions, including cardiovascular diseases, diabetic nephropathy, polycystic ovary syndrome (PCOS), and the prevention or treatment of cancer [59,72,73] (Figure 2). Hyperinsulinemia caused by insulin resistance might, in fact, induce carcinogenesis indirectly by raising the levels of steroid sex hormones and directly through the insulin receptor or insulin-like growth factors (IGF), inflammatory processes, and interrupting adipokines homeostasis [74].

## 3. Metformin as a Potential Treatment Option for Endometriosis—Mechanisms

Metformin has a pleiotropic effect by stimulating AMPK, which is the major regulator of cellular energy homeostasis that inhibits *mTOR*, the main suppressor of autophagy [55,75,76]. The activation of AMPK is of particular importance for understanding the mechanism of metformin’s action in women with endometriosis. The AMPK, as a serine/threonine-protein kinase, is a principal enzyme that regulates energy homeostasis in cells [65]. Furthermore, mitogen-activated protein kinase (MEK)/extracellular-signal -regulated kinases (ERK) phosphorylation is triggered by metformin [77,78].

Zhou et al. [65] reported that the mechanism of metformin action is the inhibition of prostaglandin E2 (PGE2)-induced cytochrome P450 19A1 (CYP19A1) mRNA expression and aromatase activity in endometriotic stromal cells (ESC) by inhibiting the binding of the cAMP-Response Element Binding protein (*CREB*) to the Proximal *Promoter*
*(**PII).* This process was associated with the activation of AMPK. It should be emphasized that profuse CYP19A1 mRNA expression and elevated local oestrogen production have been found in endometriotic tissues, implying that P450 aromatase is involved in the local oestrogen production [65]. Changes in the expression of CYP19A1 mRNA may result in a significant decrease in aromatase activity.

Additionally, it has been confirmed that metformin inhibits the mRNA expression of insulin-stimulated CYP19A1 and follicle-stimulating hormone (FSH) in granulosa cells [68,79] and also significantly decreases the forskolin/phorbol ester (FSK/PMA) -dependent upregulation of CYP19A1 mRNA expression in primary stromal cells of human breast adipose tissue [80]. The intracellular actions related to these various effects of metformin in ESCs rely on the fact that the AMPK pathway is the main mechanism of metformin action [65,80,81,82,83,84,85]. This means that metformin can reduce CYP19A1 mRNA expression and aromatase activity to some extent independently of AMPK [65].

Nevertheless, metformin regulates the expression of aromatase using other signalling pathways and unknown mechanisms. Furthermore, the AMPK pathway is gradually phosphorylated upon metformin stimulation in ESC [77]. A similar effect on the gene expression of CYP19A1 in human ESCs might have the AMPK inhibitor and PGE2 [65]. In addition, adiponectin, whose concentrations are reduced in serum and peritoneal fluid in endometriotic patients, stimulates AMPK and inhibits the production of inflammatory cytokines in endometriosis [86,87,88]. Moreover, AMPK participates in the anti-inflammatory action of metformin shown by ESCs [78]. These observations confirm that induction of AMPK in endometriosis may be a protective factor. Steroidogenesis and CYP19A1 gene expression have been shown to be regulated by the MEK/ERK signalling cascade; however, conflicting findings regarding the mechanism in various steroidogenic cells have been reported. For instance, it has been reported that inhibition of MEK activity with PD98059 and U0126 is associated with stimulation, inhibition, or no effect on the steroidogenic response [77,78,89,90,91,92]. Zhou et al. [65] demonstrated the inhibition of CYP19A1 mRNA expression by metformin as autonomous of the MEK/ERK pathway.

Stimulation of the MEK/ERK pathway may be relevant in other metformin mechanisms. Metformin may reduce PGE2-activated CREB binding to CYP19A1 PII in human ESCs [65]. Notably, metformin decreases CYP19A1 expression at the transcriptional level. It is noteworthy that metformin interacts with the FSH-stimulated cAMP/PKA/CREB pathway, being the major signalling pathway regulating the expression of the CYP19A1 gene in the ovaries [93]. The above-mentioned observations indicate that the benefits of metformin may also be mediated by the inhibition of oestrogen production in the ovaries by the same actions as outlined in ESCs.

In stromal cells of human breast adipose tissue, metformin suppresses the nuclear translocation of CREB-regulated transcription co-activator 2 (CRTC2), which elevates aromatase expression by binding to CYP19A1 PII. It is also the main purpose of AMPK [80,81]. In granulosa cells, metformin decreases the FSH-induced phosphorylation of CREB, thereby reducing the CREB action, which may cause interference of the *CREB-*binding protein *(CBP)-*CRTC2 co-activator complex that connects to CRE in PII of the CYP19A1 gene. Metformin is able to disturb the CREB-CRTC2 complex in human ESCs [77]. The above results confirm that metformin attenuates the PGE2-stimulated binding of CREB to CYP19A1 PII, possibly by interfering with the CREB-CRTC2 complex [65].

It has been confirmed that metformin interferes with the pathophysiology of endometriosis by reducing matrix MMPs expression, although it is not the major mechanism of its action [94]. MMPs are proteolytic enzymes that participate in the reduction and reconstruction of the extracellular matrix. The disturbance of the regulation of MMPs is considered crucial in the progress of pathological disorders such as endometriosis [95,96,97]. Accordingly, MMP inhibitors may impair the progression of the disease.

It has been proven that oxidative status balance may affect lipid and glucose metabolism [98]. It has also been confirmed that patients affected by endometriosis have abnormalities in both types of these metabolisms. For example, McKinnon et al. [99] reported that in endometrial lesions, the expression of the glucose-4 transporter (GLUT4) protein is increased, possibly due to elevated glucose availability in the growing lesions. Melo et al. [100] found that patients with endometriosis often have an unfavorable lipid profile (decrease in high-density lipoprotein and increase in low-density), potentially conducive to oxidative stress (by lipid peroxidation), which tends to increase in this group of women.

The unique therapeutic action of metformin in endometriosis is associated with its impact on inflammation, angiogenesis, invasion, and adhesion, as well as apoptosis. The mechanisms of action of metformin in the above-mentioned processes require a more detailed description.

### 3.1. The Impact of Metformin on Inflammation

Inflammation plays an important role in the process of endometriosis [1,101] and is associated with progesterone resistance. Many studies have revealed that metformin suppresses inflammation. In endometrial stromal cells, metformin may limit the inflammatory process by affecting the secretion of IL-6 and IL-8. Metformin inhibits the production of IL-1β and suppresses the production of IL-8. Moreover, human ESCs cultures incubated with metformin show statistically significantly reduced IL-1β-induced IL-8 production in a dose-dependent fashion [102]. The same phenomenon was found in the ability of stromal cells to transform androstenedione into estrone (reaction dependent on aromatase activity) as well as in deoxyribonucleic acid (DNA) synthesis (a marker of cell proliferation). The anti-proliferative effect of metformin on IL-8 depends on the cell type because it does not suppress the secretion of IL-8 from eutopic (normal) endometrial stromal cells.

Metformin may reduce serum levels of TNF-α in women with endometriosis [103]. Treatment with metformin can intensify the autophagy process by inhibiting mTOR [104,105]. Co-therapy of quercetin and metformin significantly inhibits mTOR messenger ribonucleic acid (mRNA) expression and increases the gene expression of autophagy factors in ectopic endometrial tissues. This combined treatment regressed endometrial implants in rodents mostly through antiestrogenic and anti-inflammatory functions.

### 3.2. The Impact of Metformin on Angiogenesis

Angiogenesis is a physiological phenomenon of blood vessel formation on the basis of the existing ones. This process requires an equilibrium between stimulatory and inhibitory signals. Once the balance is disturbed, the vascularization could be activated. Angiogenesis plays an important role in the neoplastic development and metastases [106,107]. In vitro and in vivo studies revealed that metformin could inhibit tumour angiogenesis in different mechanisms, such as the AMPK/mTOR pathway activation, downregulation of platelet-derived growth factor B (PDGF-B), and inhibition of several angiogenic-related proteins, such as vascular endothelial growth factor (VEGF), hypoxia-inducible factor-1 (HIF-1), insulin-like growth factor binding protein-2 (IGFBP-2), platelet-derived growth factor-AA (PDGF-AA), VEGF, angiogenin, matrix metalloprotein-9 (MMP-9), and endostatin [108,109,110,111,112]. In vivo, metformin limited the microvascular tumour density and modified the perivascular/endothelial cell ratio [113].

Although endometriosis is a benign condition, it is linked to various cancer-related processes including angiogenesis [114,115]. It has been reported that angiogenesis resulting in the dysfunction of the vascular basal membranes, as well as the surrounding extracellular matrix (ECM) [116], is a hallmark of endometriosis. Unfortunately, precise angiogenesis mechanisms in this disorder are still unknown [114,117]. The use of metformin prevents the progression of endometrial lesions. Metformin suppresses the proliferation of endothelial cells and reduces vasculature lesions by inhibiting the expression of VEGF. Yilmaz et al. noticed that in rats treated with metformin, the concentrations of superoxide dismutase (SOD) and MMP-2 were increased, whereas the levels of VEGF and MMP-9 were decreased in the endometrial lesions [118].

It is worth noticing that metformin decreased the risk of endometrial hyperplasia by reducing the expression levels of urothelial cancer associated 1 (UCA1), transforming growth factor-β (TGF-β) and protein kinase B, while increasing the levels of microRNA-144 and active Caspase-3 [119].

### 3.3. The Impact of Metformin on Adhesion and Invasion

Metformin may inhibit the vascular adhesion molecule expression in endometriosis. Metformin may affect the binding of monocytes to endothelial cells by reducing vascular cell adhesion molecule-1 (VCAM-1) expression on activated endothelial cells. Studies confirmed that metformin in rat abdominal endometriosis induced reduction in weight, mean area, and volume of lesions compared to no reduction in the control group [13,69]. Histological assessment of the lesions proved that metformin therapy was related to a significant decrease in the number of epithelial cells. The risk and severity of adhesions were also lower.

Lipolysis-stimulated lipoprotein receptor (LSR) revealed itself as a recent molecular constituent of tricellular contacts that have a barrier mechanism for the cellular sheet [120]. LSR recruits tricellulin (TRIC), which is the first molecular element of tricellular tight junctions. The knockdown of LSR enhances invasion and motility of some cancer cells [121]. In endometriosis, LSR is noticed in the secretory phase of normal endometrial epithelial cells. Furthermore, LSR in cancer is reduced in association with the malignancy. The downregulation of LSR by siRNA provokes cell migration, proliferation and invasion, while TRIC transfers from the tricellular region to the bicellular region of the membrane [122]. Metformin increases LSR expression and prevents the migration and invasion of the cells activated by knockdown of LSR in those treated with siRNA or leptin in endometrial cancer cells. Shimada et al. noticed upregulation of LSR by metformin via MAPK and upregulation of LSR via MAPK, PI3K and JAK2/STAT [122].

### 3.4. The Impact of Metformin on Apoptosis

Homeostasis of tissues is modulated by apoptosis. A balance between cell proliferation and apoptosis preserves this homeostasis against cellular disturbances. In endometriosis, the reduction in cell death can lead to the progression of this disease [123,124]. The ratio of cell apoptosis is suppressed in endometrial cells [125]. Moreover, in endometriosis, the induction of the NF-κB pathway is associated with both apoptosis and proliferation [126,127].

Hsieh Li et al. [128] observed that metformin reduces p53 expression levels and significantly induces apoptotic cell death. Contrary to this, Xiao et al. [129] noticed that metformin increases AMPK stimulation without modifying the expression levels of p53 and liver kinase B1 (LKB1). Metformin is able to increase LKB1 phosphorylation, promote p53 activation and AMPK, and suppress cell cycle progression [130]. Furthermore, Chen et al. [131] demonstrated that cytotoxicity induced by metformin occurs through the induction of the caspase-dependent apoptotic signalling pathway.

Metformin has inhibitory actions on cell proliferation, metastasis, apoptosis, angiogenesis, and chemoresistance in diverse malignancies in vivo and in vitro, such as ovarian, hepatocellular, and endometrial cancers [132,133,134]. The above inhibitory properties of metformin are associated with the induction of the PI3K/AKT/mTOR signalling pathway [132,133,134]. Metformin also suppresses tumour growth by stimulating the ATM serine/threonine kinase/AMPK/p53 signalling pathway and reducing the AKT/mTOR/eukaryotic translation initiation factor 4E-binding protein 1 signalling pathway. It leads to an increased response to radiation [135]. Moreover, clinical studies confirmed that metformin use is associated with higher overall survival in the group of patients with various cancers [136,137]. Metformin, by acting on the AMPK/p53 and PI3K/AKT/mTOR signalling pathways, may induce apoptosis and cell cycle arrest.

To summarize, metformin can reduce angiogenesis, inflammation, invasion, and adhesion and may cause regression of endometrial lesions. Insulin sensitizers are reported to diminish cytokine and chemokine expression in endometriotic stromal cells [138], regulate angiogenesis [53], and provoke apoptosis in endometrial lesions [57].

## 4. Metformin and Estrobolome

Dysbiosis, which is an imbalance of the gut microbiota, may have pathological consequences. This disorder disturbs homeostasis by reducing bacterial variety and enlarging the Firmicutes/Bacteroidetes ratio, which causes an inflammatory reply and metabolic status that is harmful to the epithelial of intestines [139]. Dysbiosis has been shown to affect the integrity of the intestinal epithelial barrier by reducing intercellular junctions, increasing permeability, followed by translocation of bacteria [140]. The displacement of the bacteria can lead to systemic inflammation, which inducts disease or causes its exacerbation [141].

The estrobolome is the repertoire of intestine microflora genes that can metabolize oestrogens [141]. The microbiome of the intestines significantly influences the host’s oestrogen levels by secreting β-glucuronidase, an enzyme that deconjugates oestrogen. Only unbound oestrogens are able to connect to their receptors and influence women’s physiology or clinical implications. This process allows it to connect to oestrogen receptors and leads to its further physiological effects and clinical implications [142,143,144]. On the other hand, oestrogens also affect the gut microbiome.

Estrobolome can also induce hyperestrogenic pathologies through increased numbers of bacteria producing β-glucuronidase, causing elevated levels of circulating oestrogens. These hormones stimulate the proliferation of the epithelium in the female reproductive system and have been proven to lead to proliferative disorders, including endometriosis, uterine fibroids, and endometrial cancer [145]. Endometriosis is frequent in premenopausal females and, combined with its hyperproliferative state, suggests that the disease could be aggravated by elevated oestrogen concentrations [41]. Therefore, it is promising that oestrogen-mediation may reduce the hyperproliferation associated with endometriosis. In a study of primates with endometriosis, changes in the intestine’s microbiota (lower concentrations of lactobacilli and higher levels of Gram-negative bacteria) have been shown, although the mechanisms linking these phenomena remain unclear [146]. Interestingly, gonadotropin-releasing hormone (GnRH) agonists affect local uterine microbiota, demonstrating the ability of hormonal regulation to modulate microflora composition [46]. Data explaining the activity of β-glucuronidase in the intestine microbiota of patients with endometriosis may explain the role that estrobolome plays in endometriosis. Patients with this condition may have an increased number of bacteria that produce β-glucuronidase in the gut microbiome, which can lead to elevated levels of oestrogen metabolites and thus initiate endometriosis [47].

The consequences of dysbiosis lead to ameliorating faecal microbiome (FMT) transplants, bariatric surgery, and pharmacological treatment (such as metformin). These procedures are designed to restore homeostasis by increasing the variety of the gut microbiome, reducing inflammation, and altering metabolite composition [147,148]. Metformin therapy may alleviate the associated disease state by modulating the composition of the intestinal microflora, as it has been shown to alter the composition of the intestinal microbiome, increasing the abundance of Akkermansia spp. [149].

The modulation of the gut microbiome and metabolic profiles via treating oestrogen-dependent diseases with metformin seems to be an interesting pharmaceutical option. It also shows promising results for preventing the metabolic consequences of endometriosis.

## 5. The Use of Metformin as Pharmacological Therapy in Endometriosis—Clinical Research

Metformin is a multipotential medicament with a rich history and encouraging prospects for the future. The biguanide derivative was used for the first time in the 1960s and registered as a hypoglycaemic substance in the pharmacotherapy of diabetes mellitus type 2. Metformin looks similar to a drug with one-off therapeutic potential. Clinical and experimental studies show that metformin might also be beneficial in other diseases such as polycystic ovary syndrome, cancer, or endometriosis.

In 2010, Yilmaz et al. presented the action of metformin in reducing endometrial changes in their work. First, endometriosis was induced in rats by surgery. Next, the 2 groups (group A and B) were given metformin in doses of 25 and 50 mg/kg/day, respectively, for 28 days. Group C, the control group, was given a placebo (saline). Histologic score, weight, and mean volume of implants in groups A and B were significantly lower in comparison to the control group. The activity of MMP-2 and SOD were significantly higher than in the control group. Additionally, the levels of VEGF and MMP-9 in endometrial implants were significantly decreased [121].

Xu et al. analysed and described the molecular and cellular mechanism by which metformin modulates steroidogenic acute regulatory protein (StAR) expression in human ESCs. They showed highlighted data of CREB-regulated transcription co-activator 2 (CRTC2) role in the mechanism by which metformin downregulates StAR expression. The nuclear translocation of CRTC2 is achieved by increasing AMPK phosphorylation, which inhibits transcription of StAR by blemishing the formation of CREB-CRTC2 complex involving the activation of the StAR promoter cyclic adenosine monophosphate (cAMP) response element [77].

In another study, Takemura et al. assessed the effect of metformin on the proliferation of ESCs, oestradiol production, and inflammatory response. They decided to measure IL-8 production, mRNA expression, 5-bromo-2′-deoxyuridine incorporation, and aromatase activity in ESCs using 10 μm-, 100 μm-, and 1000 μm- doses of metformin. They obtained endometriotic tissues from women who underwent surgery for ovarian endometriomas. This research revealed that metformin significantly decreased aromatase activity in ESCs, cAMP-induced mRNA expression, 5-bromo-2′-deoxyuridine incorporation, and the interleukin 1β (IL-1β)-induced IL-8 production in a dose-dependent manner with maximal effect at 1000 μm [102].

Metformin may also play a role as an anti-angiogenic agent. In 2021, Yari et al. used three types of human cells: normal ESCs (N-ESCs) from healthy endometrial tissue, eutopic ESCs (EU-ESCs), and ectopic ESCs (ECT-ESCs) from women with endometriosis. All types of cells were cultured and treated with different metformin concentrations for 72 h. The scientists assessed metformin’s effect on cell migration, proliferation, and viability. In addition, the expression of inflammatory and angiogenic genes was also checked. The results showed that metformin inhibited the proliferation and migration of cells in a concentration-dependent manner in ESCs. Moreover, metformin significantly lowered VEGF-A expression in EU-ESCs and ECT-ESCs and hypoxia-inducible factor 1α (HIF-1α) in ECT-ESCs. EU-ESCs and N-ESCs presented a significantly decreased macrophage migration inhibitory factor (MIF) expression after metformin treatment. Although EU-ESCS showed decreased gene expression of MMP-2 and MMP-9 after metformin use, in ECT-ESCs and N-ESCs, the expression of tissue inhibitor of MMPs was significantly increased after drug treatment [150].

Jamali et al. analysed the effects of quercin, metformin, and their combination on experimental endometriosis in a rat model. A total of 60 female rats were divided into six groups. Five of them underwent surgery to induce endometriosis; after 4 weeks, daily treatment lasting 4 weeks began. Afterward, histoarchitecture and size of the endometrial implants, serum levels of progesterone, TNF-α and 17β-estradiol, markers of autophagy, and oxidative stress were assessed using gene expression analysis and enzyme-linked immunosorbent assay (ELISA). The results showed that serum levels of TNF-α and 17β-estradiol were higher in rats with endometriosis. Additionally, quinone oxidoreductase (NQO1) enzyme reduced nicotinamide adenine dinucleotide phosphate (NADPH) activity and gene expression levels of autophagy markers, and nuclear factor erythroid 2-related factor 2 (Nrf2) were significantly decreased. Moreover, mTOR gene expression was higher in the ectopic endometrial tissue in comparison to the eutopic one. The combination of quercin and metformin reversed the alterations and had a marked effect on the size of endometrial implants and gene expression levels of autophagy markers and mTOR in ectopic endometrial cells [55].

Oner et al. investigated the effects of letrozole and metformin on experimentally induced endometriosis in rats. A total of 38 female rats were divided into four groups: a control group, two metformin groups, and a letrozole group. Rats were treated for 4 weeks; next, they were sacrificed, and the size of endometrial implants and scores of adhesions were assessed. The surface area of endometrial implants was significantly reduced in treatment groups, and the effect was comparable in metformin groups and letrozole groups. The histopathologic assessment indicated that the histopathologic score was lowest after 100 mg/kg/day dose of metformin. Moreover, metformin decreased the severity of adhesions [69].

Zhou et al. analysed the influence of metformin on PGE2-induced cytochrome P450 aromatase activity via the stimulation of AMPK and gene expression in human ESCs. In the research, ESCs were cultured by PGE2, AMPK inhibitors. Expression of *CYP19A1* mRNA and aromatase activity was measured by aromatase activity assay and qPCR, respectively. The binding of CREB protein to CYP19A1 PII was estimated by chromatin immunoprecipitation (ChIP) assay. The results showed that metformin decreased the expression of aromatase mRNA and activity stimulated by PGE2 in ESCs via induction of AMPK. After PGE2 treatment, there was a pronounced increase in CREB binding to aromatase PII and an inhibition of the PGE2-induced stimulation by metformin [65].

Zhang et al. studied the effect of metformin on stromal-epithelial cells crosstalk in endometriosis. The communication via Wnt2/β-catenin signalling in endometriosis was assessed. Their results showed that ESCs expressed and secreted increased levels of Wnt2 protein in comparison to normal endometrial stromal cells (NSCs). Recombinant Wnt2 significantly enhanced the expression of β-catenin in normal endometrial epithelial cells (NECs), while the Wnt2 antibody exerted no statistically significant effect on the expression of β-catenin in NECs. However, the supplementation of Wnt2-antibody significantly raised the inhibition of β-catenin expression by conditioned medium (CM)-ESCs in NECs. After metformin treatment, the secretion and expression of Wnt2 protein in ESCs were decreased substantially. CM from metformin-pre-treated-ESCs significantly reduced the growth of NECs [151].

## 6. Possible Beneficial Indications for Using Metformin in Endometriosis

The use of metformin may be recommended in endometriotic patients with contraindications to hormonal therapy, in patients with side effects to medical therapy, in patients resistant to other treatments, or in women who desire the chance to conceive.

Metformin can be used as a co-therapy with hormones, enabling a reduction in their dosage. In infertility related to endometriosis, the use of anti-endometrial drugs could be favourable for the induction of ovulation. Aromatase inhibitors are efficient in treating infertility; however, they produce hypoestrogenic effects [152]. Metformin therapy has no serious side effects and may therefore be more beneficial. Its anti-estrogenic, anti-inflammatory, and anti-proliferative potential makes it a therapeutic option in endometriosis [153].

Metformin therapy can significantly reduce a patient’s symptoms. Metformin is able to suppress endometriosis by inhibiting both ovarian and local oestrogen production [138,153]. Metformin therapy profoundly decreases the incidence of symptoms linked to endometriosis, including pelvic pain, dysmenorrhea, and dyspareunia. Furthermore, metformin has a beneficial effect on ovulation in comparison to other treatments that do not provide a chance at conception. These results suggest an exceptional therapeutic potential for metformin in managing endometriosis in patients who desire pregnancy.

Endometriosis is the direct cause of adhesion formation [72]. Endometriosis-related inflammation, through increased concentrations of VEGF in the peritoneal fluid, may provoke angiogenesis for the progressive growth of endometriosis [68]. Early diagnosis and treatment with tolerable side effects are essential, especially in young patients. This management could prevent disease progression.

Importantly, metformin eases weight loss; thus, it has an indirect potential on oestradiol. The decrease in fat tissue suppresses endogenous oestradiol and positively affects endometriosis implants [68]. Advanced endometriosis is more common in patients who suffer from obesity [154]. Obesity is linked to hyperestrogenism. This aspect is particularly significant for patients with endometriosis who are undergoing infertility treatment. Obese women may experience the greatest benefit in this area. In addition, the live birth rate is enhanced in patients who received metformin [155].

To summarise, the medical management of endometriosis by metformin focuses on the improvement of a patient’s quality of life by reducing the size of implants, lesions, and pelvic pain. Additionally, it helps in preserve fertility by improving pelvic organs’ function.

## 7. Side Effects of Using Metformin

Though there are benefits to its use, metformin therapy has several side effects. Some of the most common side effects associated with the use of biguanide derivatives are gastrointestinal symptoms [156,157]. Due to intestinal accumulation of the substance, approximately 20–30% of patients may manifest mild to moderate ailments [158]. In the study conducted by Ma et al. 40% of patients suffered from nausea. Another adverse reaction was diarrhoea. Other symptoms include abdominal distension, constipation, vomiting, abdominal discomfort or indigestion [159]. Within the first 8 weeks of therapy, patients may feel fatigue and/or complain about headache. It is estimated that about 5% of the population treated with metformin develop severe complications such as memory impairment, cardiovascular instability (prolonged QTc interval) or acute kidney injury [160,161]. An impaired cognitive function is mostly connected with B12 vitamin deficiency that develops due to malabsorption [162,163]. An extremely rare, life-threatening state is lactic acidosis entailing irregular heartbeat, difficulty breathing, dizziness, tiredness, stomach pain with significant diarrhoea [164]. Chronic diarrhoea, that requires hospitalization, also manifests as symptomatic electrolyte abnormalities such as hypocalcaemia, hypokalaemia, hypomagnesemia, and hypophosphatemia. Most gastrointestinal symptoms can be alleviated by supplementation with fibre or vitamin B12 [165].

## 8. Conclusions

The reduction in endogenous steroid production is the basis for the treatment of endometriosis. Nevertheless, sustained use of medicaments including progestogens, GnRH-analogs, danazole, and aromatase inhibitors have limitations connected to their side effects, cost, delayed conception, limited treatment time, and the high rate of recurrence after discontinuation. There is also no advantage in their use in patients with endometriosis-related infertility [52,166].

Metformin is a drug generally available, cheap, and easy to use in oral therapy. This well-tolerated medicament relieves pain, alleviates menstrual disorders, and improves fertility in endometriosis. Metformin therapy in endometriotic patients is associated with a statistically significant limitation in the symptomatic cases, enhanced chance of conception, and suppression of the concentrations of cytokines in serum, suggesting that it may have a beneficial effect as an anti-endometriotic drug [167].

Published studies assessing the effect of metformin in endometriosis are promising. Studies based on in vitro and animal models confirm that metformin is associated with the recession of endometriotic implants [50]. Notwithstanding, the number of available studies is still inadequate on the consequences of metformin use in endometriosis. Unfortunately, the results of the studies are varied, and clinical trials are rare. Therefore, further research on the effects of metformin as an anti-endometriotic drug is needed. Prospective multicentre clinical trials would be invaluable to confirm the beneficial potential of metformin in the treatment of endometriosis.

## Figures and Tables

**Figure 1 cancers-14-00577-f001:**
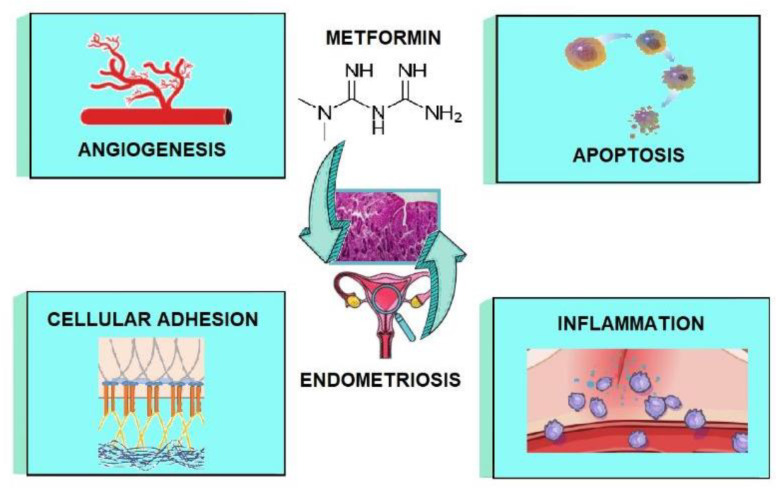
The potential actions of metformin in endometriosis.

**Figure 2 cancers-14-00577-f002:**
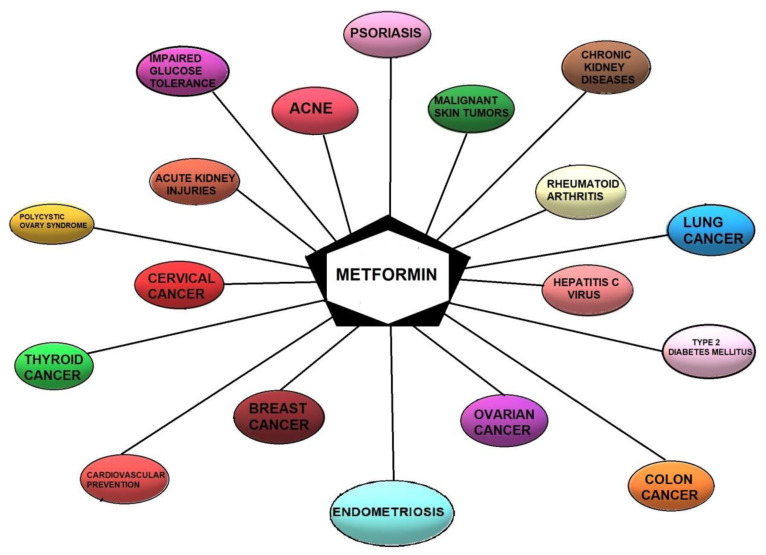
Current and potential therapeutic indications for metformin.

## Data Availability

MDPI Research Data Policies.

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
