# Peer review of "Metformin as a Potential Treatment Option for Endometriosis"

_cancers, 2022, doi:10.3390/cancers14030577_

Round 1

Reviewer 1 Report

"I applaud the Authors for the article.

Please better discuss the side effects of metformin.

I suggest to include some relevant papers to elucidate the hypoxia in
DIE nodules (i.e. DOI 10.1016/j.fertnstert.2018.02.122).

Please underline the goals of the medical management for patients affected
by endometriosis: QoL improvement, reduction of pain and pelvic organs'
function and preservation of fertility.”

Reviewer 2 Report

The topic of the review is very interesting and relevant, the possibility of treating endometriosis with metformin is now being actively investigated.The review is devoted to the possible mechanisms of action of metformin in this pathology.Much attention has been paid to various signaling pathways, the effect of the drug on which can determine the multiple beneficial effects of metformin.

Unfortunately, the authors clearly do not cope with the systematization and presentation of a large amount of information.The general outline is good, but the content does not always correspond to the headings, the text is illogical, and some links are inadequate.In many places, it is difficult to read and understand the meaning of the text, since the information comes down to a very general mention of signaling pathways and other regulatory mechanisms, or rather strange facts are mentioned, without links and explanations.

For example:

Line 58: "MMPs indicate that maintenance and survival of endometrial lesions is possible by modulating the expression of progesterone." The relationship between progesterone and the growth of endometriosis foci is quite complex, but it is not very clear what does MMPs have to do with it here.

  In many places, words are too general: Line 65: "The COX-2 / PGE2 pathway increases production of estrogen in the endometrium by acting on the G protein-coupled receptors." The G protein-coupled receptors are somewhat vague, as is the tyrosine kinase activity at line 170.

Professionals usually do not decipher the names of genes and proteins (BCLXL on line 79 and mTOR on line 198) - this is absolutely meaningless in this context. And by the way, the article at reference 27 does not contain any information that "PI3K / Akt pathway ... controls the B cell lymphoma extra-large (BclxL) expression" (line 79) 

At the beginning of Section 3 - "Metformin as a Potential Treatment Option for Endometriosis - Mechanisms", in fact, the different ways in which metformin affects aromatase activity are discussed.

Subsection 3.1. “The Impact of Metformin on Inflammation” has one paragraph on the LSR receptor, which is more related to adhesion / invasion. The phrase on line 301 "Angiogenesis is a dysfunction of the basal membrane of the vessels and the surrounding extracellular matrix (ECM)" is beyond good and evil. And this whole section is written in a peculiar way. It's very short. : “Angiogenesis is a dysfunction of the basal membrane of the vessels and the surrounding extracellular matrix (ECM). The use of metformin prevents the progress of endometrial lesions. Metformin suppresses the proliferation of endothelial cells and reduces vasculature lesions by inhibiting the expression of VEGF. Yilmaz et al. noticed that in the lesions of the treated rats the concentrations of superoxide dismutase (SOD) and MMP-2  were increased, and that the levels of VEGF and MMP-9 were decreased [109] "

And so on. This is a very low quality of text for a highly rated journal.

Reviewer 3 Report

The paper entitle: “Metformin as a potential treatment option for endometriosis.”  The author organized many references and demonstrated metformin may be a potential treatment of endometriosis. The author reviewed the current data about the metformin on the endometriosis. First, they showed the endometriosis and its treatment. Second, the author reviewed metformin and its biological functions and effects.

The reviewer thinks this manuscript is organized and discussed. However, the writing is not sophisticated to the reader. There are many grammar errors and typos. Please check it carefully throughout all manuscript. An English speaking person should be asked to correct the language.

  1. In Figure 2, large intestinal cancer should be colon cancer.
  2. In abstract, line 28. What is the implant levels of superoxide dismutase tissue ?
  3. line 45. One of the latest, oft-cited and widely presented theories postulates that endometriosis arises from an abnormal response to inflammatory process. What is oft-cited ?
  4. line 56. remodeling of the cellular matrix is supervised by MMP-1, 2, 3, 9

It should not be use ‘supervised’

  1. line 59. What is more, MMP-2 and 59 MMP-9 control the cyclooxygenase-2 (COX-2) sensitivity in the endometrial cells

What is “What is more,” ?

  1. line 106. Therefore it is considered as an insulin sensitizer.

  This sentence lacks one comma.

  1. Line 180. What is AMPK. The full name should be listed the first time.
  2. Line 207. What is aroPGEase ?
  3. line 249 There are two CREB
  4. Line 250. Metformin is able disturb to CREB-CRTC2 complex in human ESCs.

   There is a missing “to”

  1. Line 276-282. It is hard to understand.

   human ESCs cultures incubated with metformin reveal a reduction 278 in IL- β1-induced IL-8 production [101].

   There are three verbs in one sentence.

   Moreover, line 278 showed that metformin decreased IL-8, but it is not suppressed in line 282. What is the conclusion of the regulation of IL-8 by metformin ?

Round 2

Reviewer 2 Report

The authors have made all the recommended corrections. Perhaps, in the section "Pathophysiology of Endometriosis" it would be worth mentioning other theories of the occurrence of this disease, in addition to the theory of retrograde menstruation, the plausibility of which will now raise many questions. I also expected to read in this paper about metabolomic disorders in patients with endometriosis, but this issue is also not considered. Nevertheless, the review corresponds to the stated topic and may well be published. 

Reviewer 3 Report

The authors have revised the manuscript.